# Goal-oriented care for patients with chronic conditions or multimorbidity in primary care: A scoping review and concept analysis

Dagje Boeykens[1,2☯]*, Pauline Boeckxstaens[2☯], An De Sutter[2], Lies Lahousse[3], Peter Pype[2,4], Patricia De Vriendt[1,5,6‡], Dominique Van de Velde[1,5‡], on behalf of the Primary Care Academy[¶]

1 Faculty of Medicine and Health Sciences, Department of Rehabilitation Sciences, Occupational Therapy, Ghent University, Ghent, Belgium, 2 Faculty of Medicine and Health Sciences, Department of Public Health and Primary Care, Center for Family Medicine, Ghent University, Ghent, Belgium, 3 Faculty of Pharmaceutical Sciences, Department of Bioanalysis, Ghent University, Ghent, Belgium, 4 Faculty of Medicine and Health Sciences, End-of-Life Care Research Group, Vrije Universiteit Brussel and Ghent University, Ghent, Belgium, 5 Department of Occupational Therapy, Artevelde University of Applied Sciences, Ghent, Belgium, 6 Faculty of Medicine and Pharmacy, Department of Gerontology and Mental Health and Wellbeing (MENT) Research Group, Frailty in Ageing (FRIA) Research Group, Vrije Universiteit Brussel, Brussel, Belgium

☯ These authors contributed equally to this work.
‡ These authors also contributed equally to this work.
¶ Membership of the Primary Care Academy is provided in the Acknowledgments.
* Dagje.boeykens@ugent.be

**Data Availability Statement:** All relevant data are within the paper and its Supporting Information Files.

## Abstract

### Background

The healthcare system is faced by an ageing population, increase in chronic conditions and multimorbidity. Multimorbid patients are faced with multiple parallel care processes leading to a risk of fragmented care. These problems relate to the disease-oriented paradigm. In this paradigm the treatment goals can be in contrast with what patients value.

The concept of goal-oriented care is proposed as an alternative way of providing care as meeting patients' goals could have potential benefits. Though, there is a need to translate this concept into tangible knowledge so providers can better understand and use the concept in clinical practice. The aim of this study is to address this need by means of a concept analysis.

### Method

This concept analysis using the method of Walker and Avant is based on a literature search in PubMed, Embase, Cochrane Library, PsychInfo, CINAHL, OTSeeker and Web of Science. The method provides eight iterative steps: select a concept, determine purpose, determine defining attributes, identify model case, identify additional case, identify antecedents and consequences and define empirical referents.

### Results

The analysis of 37 articles revealed that goal-oriented care is a dynamic and iterative process of three stages: goal-elicitation, goal-setting, and goal-evaluation. The process is

**Funding:** D.B. is a PhD student payed by the King Baudouin Foundation. Grant number: 2019-J5170820-211588 - King Baudouin Foundation - https://www.kbs-frb.be/nl/ - The funders had no role in study design, data collection and analysis, decision to publish, or preparation of the manuscript.

**Competing interests:** The authors have declared that no competing interests exist.

underpinned by the patient's context and values. Provider and patient preparedness are required to provide goal-oriented care. Goal-oriented care has the potential to improve patients' experiences and providers' well-being, to reduce costs, and improve the overall population health. The challenge is to identify empirical referents to evaluate the process of goal-oriented care.

## Conclusion

A common understanding of goal-oriented care is presented. Further research should focus on how and what goals are set by the patient, how this knowledge could be translated into a tangible workflow and should support the development of a strategy to evaluate the goal-oriented process of care.

## Introduction

The healthcare system is faced by an ageing population and an increase in chronic conditions and multimorbidity [1]. More and more people are forced to live with the consequences of these demographic changes and require ongoing (chronic) care on top of acute care [2]. At the same time, patient autonomy is gaining importance and patients are considered as an active and important partner in their care [3,4]. Patients with chronic conditions are often consulting multiple health care providers [3] leading to a higher rate of encounters. They also receive a larger amount of prescriptions [5] and they are asked to complete a diverse set of self-monitoring tasks such as managing, exacerbations or monitoring biomedical targets [3]. Since patients with (multiple) chronic conditions are faced with multiple parallel care processes for their different conditions, there is a considerable risk of fragmented care. Especially when healthcare providers focus on disease control, patients can experience lack of care continuity and issues with communication as patients themselves focus on the meaning of care and more on personal wellbeing [6,7]. As a result, treatment goals can be in contrast with what patients value in their personal lives [3].

The healthcare system is oriented towards a disease-oriented paradigm to which many of these problems relate [8–10]. In this paradigm, care is mainly organized according to disease-oriented guidelines [10]. This may work well for patients with a single disease, but becomes inappropriate for patients with multiple problems. The focus on single disease guidelines might distract providers from what really matters to the patient [10]. A possible way to overcome many of the challenges is to shift care back from 'what's the matter with the patient' to 'what matters to the patient'. It creates healthcare processes in which patients' needs are actively sought and met [9]. Meeting those patients' needs and tailoring care more to what patients want in a co-creation process could result in better social well-being, physical well-being, and satisfaction for patients and healthcare providers [11].

One of the possible strategies is to actively engage patients in identifying their personal goals and aligning care to those goals, which could be achieved by goal-oriented care [12]. The concept of goal-oriented care was conceived for the first time in 1991 by Mold who proposed the concept as an alternative way of providing care [13]. Later on, in 2012, Reuben and Tinetti took the concept of goal-oriented care a step forward by stating that care "must above all consider patients' preferred outcomes" [10]. The focus on setting goals based on the patients' needs and preferences rather than on health-related outcomes became one of the main novelties in chronic disease management [4]. Not only could goal-oriented care be proposed as an

important paradigm to overcome some of the new challenges for chronic patients [9], it might also corresponded to the original concept of evidence based medicine (EBM) [14]. EBM was described by Sackett in 1996 who presented three key components: 1. best external evidence, 2. individual clinical expertise, and 3. patients' values and expectations [14]. Since the first description of EBM, multiple approaches and paradigms has been developed to compromise between those three components [15]. For example, patient-centered care (PCC), which is already a well-known and widely used concept, is defined as "providing care that is respectful of, and responsive to individual patient preferences, needs, and values and ensuring that patients values guide all clinical decisions" [15]. Shared-decision making, on the other hand, also strives to share evidence and engage patients in care as it is "an approach where clinicians and patients share the best available evidence when faced with the task of making decisions, and where patients are supported to consider options, and to achieve informed preferences" [16]. Goal-oriented care is proposed as a promising healthcare paradigm and approach to operationalize EBM and return to where it all started [10]. However, in contrast to the other approaches and paradigms, goal-oriented care is ill defined. Developing a common under-standing on the concept could potentially contribute to the clarification and in-depth compari-son between the related concepts and eventually lead to better use in clinical practice. However, some healthcare providers might already assume that they practice goal-oriented care spontaneously, but there still is a lack of underpinning knowledge and guidance on how to provide goal-oriented care to patients. The main pitfall in most of these goal-setting activi-ties is that the goals are not necessarily related to the patients' needs and preferences while in goal-oriented care these patients' needs and preferences are put on the forefront and are not necessarily health-related [17,18]. From this perspective, goal-setting and goal-oriented care should be taken together and focus on the patients' needs and preferences.

As a first step in exploring the potential of goal-oriented care in chronic care, it is important to gain in-depth knowledge on what goal-oriented care is about and how it can be generally described.

As goal-oriented care could be well-suited in primary care, as this context is often the linch-pin for patients with chronic conditions, this is the focus of this study [19]. This study aimed to describe a structured approach to deepen the concept of goal-oriented care for patients with chronic conditions or multimorbidity in the primary care context.

## Method

This concept analysis aims to present an overview and synthesis of the existing literature regarding goal-oriented care for chronically ill patients in primary care. This will be performed by analyzing the concept into antecedents, attributes, and consequences following the method of Walker and Avant [20]. This method provides a framework of eight iterative steps: 1. select a concept, 2. determine the aims or purposes of analysis, 3. identify all concept definitions and select the literature, 4. determine different attributes, 5. identify a model case, 6. identify an additional case, 7. identify antecedents and consequences, and 8. define empirical referents [20]. In this concept analysis the attributes are the heart and will present the characteristics of goal-oriented care and allow the broadest insight into the concept [20].

### Step 1: Select a concept

Goal-oriented care has been defined as an underpinning strategy for primary care reform in Flanders, Belgium. The concept is presented as one of the main topics of 'The Primary Care Academy' (PCA). The PCA is a consortium consisting of four universities (Ghent University, University of Antwerp, Catholic university of Leuven, Vrije Universiteit of Brussels), six

universities of applied sciences (UAC VIVES, UAC Artevelde, UAC Ghent, UAC Leuven-Limburg, UAC Karel de Grote, UAC Thomas More), and important stakeholders (Flemish Patient Platform and White-Yellow Cross; a home care organization) in Belgium with the aim to strengthen the organization and delivery of primary care. The PCA includes experts in primary care from a variety of healthcare and welfare disciplines. Discussions in the research group working on goal-oriented care created a necessity to clarify the concept.

## Step 2: Determine the aims and purposes of the analysis

The aim of this concept analysis is to build a common understanding to eliminate ambiguity between the concepts related to goal-oriented care. Specifically, the scope of the concept analysis is to define goal-oriented care for people with chronic conditions at the level of primary care.

## Step 3: Select the literature

The literature was searched between January 2020 and April 2020. As the method of a concept analysis does not specify how the literature search has to be performed, this search was based on the method of a scoping review described by Levac (2010) [21]. A preliminary combination of search terms was identified: 'goal-oriented care', 'chronic care', and 'primary care'. Based on these keywords a first search was performed to identify adjacent terms in the literature. The search strategy was revised in consultation with the librarian of the university and the senior researchers. The definitive keywords were: 'goal-oriented care', 'goal-oriented medical care', 'person-centered goal-setting', 'patient-centered goal-setting', 'goal-oriented patient care', and 'patient priorities', emphasized goal-oriented care and it synonyms. Related concepts such as patient-centered care, value-based care, etc. were not included as the method of concept analysis prescribes to deepen all the attributes of one concept. In a first phase, the keywords were entered in PubMed, Embase, and Cochrane Library (Table 1). In a second phase, CINAHL, OTSeeker, PsycINFO, and Web of Science were consulted and confirmed the first results as no new studies were identified.

Articles resulting from this search were put in Rayyan [22] to administer the data. A first selection based on title and abstract was performed with regard to the predefined in- and exclusion criteria. Inclusion criteria: (a) goal-oriented care as a health-related concept, (b) mentioning goal-setting, goal-oriented care or related concept (e.g., person-centered integrated care), and (c) focusing on patients with one or more chronic conditions. Exclusion criteria: (a) focusing on single-disease management, (b) goals regarding disease-specific outcomes (e.g., cancer or diabetes), (c) focusing on goal-oriented care in a specific context (e.g., rehabilitation center), and (d) specifically mentioning patient-centered care, shared-decision making, etc. as they will hamper the understanding of specifically goal-oriented care.

**Table 1. Overview of the search strings.**

| |
|---|
| **PubMed** |
| (goal-directed care[MeSH Terms]) OR goal-oriented care [Title/abstract]) OR goal-oriented medical care [Title/abstract]) OR person-centered goal-setting [Title/abstract]) OR patient centered goal-setting [Title/abstract]) OR goal-oriented patient care [Title/abstract]) OR patient priorities [Title/abstract]) |
| **Embase** |
| **'goal-oriented care'**:ab,ti OR **'goal-oriented medical care'**:ab,ti OR **'person-centered goal-setting'**:ab,ti OR **'patient centered goal-setting'**:ab,ti OR **'goal-oriented patient care'**: ab,ti OR **'patient priorities'**:ab,ti |
| **Cochrane** |
| goal-oriented care in Title Abstract Keyword OR goal-oriented medical care in Title Abstract Keyword OR person-centered goal-setting in Title Abstract Keyword OR patient-centered goal-setting in Title Abstract Keyword OR goal-oriented patient care OR patient priorities in Title Abstract Keyword—(Word variations have been searched) |

Articles resulting from this first search were subjected to a full text screening based on the initial criteria and: (a) full text available, (b) written in English, (c) referring to goal-oriented care or related concepts as a concept, and (d) containing information of a theoretical building of a definition. There was no restriction by study design to gain as much insight in goal-oriented care from different data sources.

### Step 4: Defining the attributes

The determination of the attributes started with a discussion of four key articles [1,6,23,24] selected by the first author based on the divers approaches of goal-oriented care and presented to the research group. Similar to a qualitative, thematic analysis, the key articles were analyzed based on an open coding and then grouped into codes (Table 2 –example of data analysis). These codes were then presented to and discussed with the co-authors. In these discussion rounds, codes were translated into attributes. In a second phase, new articles were added and analyzed based on the same method as the key articles until all relevant literature (based on the inclusion criteria) was included. The different codes were put into NVIVO12 to synthesize the data and to initiate further discussion with the research group. This resulted in the final attributes (Table 4). The method starting from reading the first article to defining the attributes was characterized by an iterative process in which the attributes were reformulated until consensus with the research group was reached.

### Step 5: Identify a model case, a contrary case, and a borderline case

A model case is presented as a narrative of how goal-oriented care could be conceptualized and illustrates all defined attributes of goal-oriented care [20]. A contrary and borderline case differ from this model case and do not include all of the attributes and/or differ in one of them.

### Step 6: Identify antecedents and consequences

Antecedents are events or incidents that precede the process of applying goal-oriented care. Consequences are those events or incidents as a result of applying goal-oriented care [20].

The antecedents and consequences were searched simultaneously with the attributes (step 4). Results have been discussed by the entire research group until consensus was reached.

### Step 7: Define empirical referents

Empirical referents provide an overview of the identified assessment tools related to the attributes aiming to make the concept, goal-oriented care, measurable. These assessment tools may

**Table 2. Example of analysis process of the study of Bernsten et al. 2018.**

| Extract from article | Code | Attribute |
|---|---|---|
| . . .A professional and a personal goal clashes in a decision process regarding the discontinuation of a medication the informant had been using for years. . . | Negotiation goals between professionals and patients. | Goal-setting–patient-provider interaction |
| . . . However "What matters to you?" gave a richer and more immediate insight into areas threatened by health issues. . . | Patient centeredness | Tailoring to patients' needs and preferences |
| . . .Goal evaluation serves as feedback to all contributors in the seamless care process. . . The result should be documented and linked back to goal adjustment and learning for the next cycle. . . | Feedback to the care process | Goal-evaluation |

be seen as the underpinning needs and characteristics when developing an evaluation method of goal-oriented care.

## Results

### Step 1–3

A first search based on the predefined terms (Table 1) resulted in 590 articles; 82 from Cochrane Library, 188 from Embase, and 313 from PubMed. After removing the duplicates, 366 articles were screened by title and abstract yielding 68 articles. A full text screening of these 68 articles led to 15 articles that fitted the predefined in- and exclusion criteria (step 3). Based on the snowballing method of adding new articles based on references, citations, and similar articles 22 additional articles were added. This resulted in a total of 37 articles (Fig 1 and Table 3) that were selected for the full text analysis. These articles represented a broad range of study types: 4 systematic reviews, 4 experimental studies (e.g., randomized controlled trial), 13 qualitative studies, 3 survey studies, 1 concept analysis, 1 methodology paper, 4 reviews, 2 position papers, 1 background paper, 1 status report, 1 commentary, 1 opinion paper, and 1 perspective.

### Step 4: Attributes

The systematic analysis of the 37 selected papers could identify many different attributes of goal-oriented care (S1 Table). Synthesizing these attributes, goal-oriented care could be

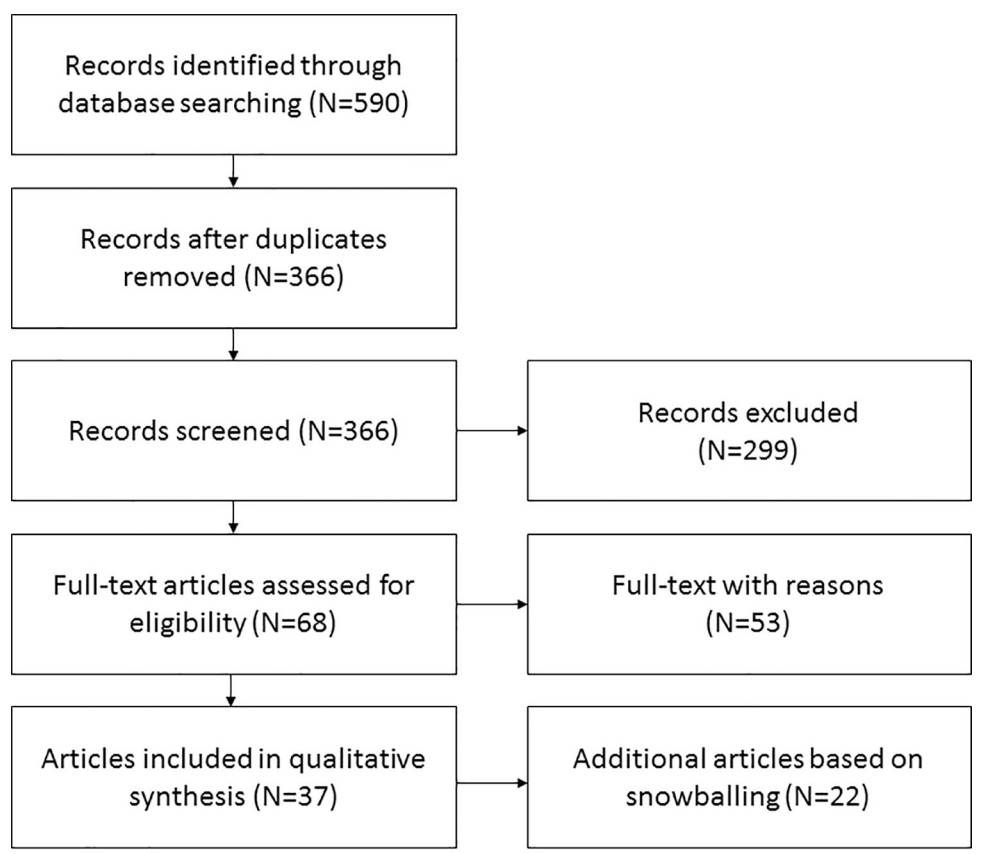

**Fig 1. Flow chart demonstrating the search string.**

**Table 3. Overview of the included articles.**

**Papers identified based on full text screening**

| No. | Year | Authors | Title | Study design + method |
|---|---|---|---|---|
| 1 | 1991 | Mold, Blake, Lorne, Becker [13] | Goal-oriented medical care. | Position paper |
| 2 | 2011 | De Maeseneer, Boeckxstaens [25] | Care for non-communicable diseases (NCD's): time for a paradigm-shift. | Opinion paper |
| 3 | 2012 | Reuben, Tinetti [10] | Goal-oriented patient care- an alternative health outcomes paradigm. | Perspective |
| 4 | 2014 | Bayliss, Bonds, Boyd, Davis, Finke, Fox, Stange [26] | Understanding the context of health for persons with multiple chronic conditions: moving from what is the matter to what matters. | Forty-five experts met to critically consider four aspects of incorporating context into research on multiple chronic conditions. |
| 5 | 2014 | Kramer, Bauer, Dicker, Durusu-Tranriover, Ferreira, Rigby, van Hulstejn [8] | The changing face of internal medicine: patient- centered care. | Position paper |
| 6 | 2015 | Bernsten, Gammon, Steinsbekk, Salamonsen, Foss, Ruland, Fonnebo [27] | How do we deal with multiple goals for care within an individual patient trajectory? A document content analysis of health service research papers on goals for care. | Document content analysis of seventy health service research papers on the topic of 'goals of care'. |
| 7 | 2016 | Blom, Elzen, Houwelingen, Heijmans, Stijnen, Van Den Hout, Gussekloo [28] | Effectiveness and cost-effectiveness of a proactive, goal-oriented, integrated care model in general practice for older people. A cluster randomized controlled trial: integrated systematic care for older people-the ISCOPE study. | Cluster randomized controlled trial–intervention group: general practitioners made an integrated care plan using functional geriatric approach; control group: care as usual; 59 general practices were included (30 intervention, 29 control); outcome measures on quality of life, activities of daily living, satisfaction with delivered healthcare, and cost-effectiveness of the intervention 1-year follow-up. |
| 8 | 2016 | Boeckxstaens, Willems, Lanssens, Decuypere, Brusselle, Kühlein, Sutter [29] | A qualitative interpretation of challenges associated with helping patients with multiple chronic diseases identify their goals. | Qualitative research–qualitative interviews with nineteen patients diagnosed with chronic, obstructive pulmonary disease and comorbidities to explore goal-setting in patients with multimorbidity. |
| 9 | 2016 | Mangin, Stephen, Bismah, Risdon [30] | Making patient values visible in healthcare: a systematic review of tools to assess patient treatment priorities and preferences in the context of multimorbidity. | Systematic review–data sources: Medline, Embase, Cochrane databases; citations were included if they reported a tool to use a record patient priorities or preferences for treatment, and quantitative or qualitative results following administration of the tool. |
| 10 | 2016 | Schimdt, Babac, Pauer, Damm, von der Schulenberg [31] | Measuring patients priorities using the Analytic hierarchy process in comparison with best-worst scaling and rating cards: methodological aspects and ranking tasks. | Analysis of the results of non-standardized Analytic Hierarchy Process (AHP)for different consistency ration threshold, aggregation methods, and sensitivity analysis; comparison of rakings criteria of AHP with best-worst-scaling and ranking cards results by Kendall's tau b. |
| 11 | 2016 | Tinetti, Esterson, Ferris, Posner, Blaum [1] | Patient priority-directed decision making and care for older adults with multiple chronic conditions. | Review |
| 12 | 2018 | Bernsten, Hoyem, Lettrem, Rul, Rumpsfeld, Gammon [6] | A person-centered integrated care quality framework, based on qualitative study of patient's evaluation of care in light of chronic care ideals. | Qualitative evaluative review of the individual patient pathways experiences of nineteen strategically chosen persons with multimorbidity. |
| 13 | 2019 | Feder, Kiwak, Costello, Dindo, Hern, Bigos, Naik [3] | Perspective of patients in identifying their values-based health priorities. | Qualitative study using in-depth semi structured telephone and in-person interviews; open-ended questions about patient perceptions of the patient health priorities identification process, perceived benefits of the process, enables and barriers to PHPI, and recommendation for process enhancement. |
| 14 | 2019 | Franklin, Lewis, Willis, Roger, Venville, Smith [32] | Controlled, constrained or flexible? How self-management goals are shaped by patient-provider interactions. | Conversation analysis; observations of consultations for chronic care management between patients and their health professionals. |
| 15 | 2019 | Tinetti, Dindo, Smith, Blaum, Costello, Ouellet, Naik [33] | Challenges and strategies in patient's health priorities-aligned decision-making for older adults with multiple chronic conditions. | Participant observation qualitative study–clinicians followed a training and had experiences in providing patient priorities care (PPC), clinicians and PPC implementation team participated in 21 case-based, group discussions. Using emergent learning, participants discussed challenges, posed solutions, and worked together to determine how to align care options with the health priorities of 35 patients participating in the patient priorities care pilot. |

**Papers identified through snowballing**

| No. | Year | Authors | Title | Study design |
|---|---|---|---|---|
| 16 | 2006 | Hurn, Kneebone, Cropley [34] | Goal setting as an outcome measure: a systematic review | Systematic review–data sources included a computer-aid literature search of studies examining the reliability, validity, and sensitivity of goal-setting/ goal-attainment scaling, with snowballing. |
| 17 | 2009 | Bodenheimer, Handley [35] | Goal-setting for behavior change in primary care: an exploration and status report. | Exploration and Status report–literature search on goal-setting interventions for promoting behavior change; resulting in eight articles. |

*(Continued)*

**Table 3.** (Continued)

| # | Year | Title | Authors | Description |
|---|------|-------|---------|-------------|
| 18 | 2011 | Health and treatment priorities of older patients and their general practitioners: a cross-sectional study. | Junius-Walker, Stolberg, Steinke, Theile, Hummers-Pradier, Dierks [36] | Cross-sectional study– 123 older patients and 11 general practitioners evaluated the importance and severity of patients' individual health problems. Patients received a geriatric assessment, then GPS rated the importance and components of severity of each problem; assessing proportion of important problems and the chance corrected agreement; multilevel logistic regression models were used to relate the importance of a problem with its severity components. |
| 19 | 2012 | Chronic disease management programs: an adequate response to patients' needs? | Rijken, Bekkema, Boeckxstaens, Schellevis, De Maeseneer, Groenewegen [2] | Survey among country-experts resulting in information about existing disease management programs; in addition scientific literature. |
| 20 | 2014 | Setting goals in chronic care: shared decision making as self-management support by the family physician. | Lenzen, Daniëls, van Bokhoven, der Weijden, Beurskens [37] | Background paper to contribute to the understanding of goal-setting within self-management and to identify elements that need further development for practical use. |
| 21 | 2016 | Supporting goal-oriented primary health care for seniors with complex care needs using mobile technology: evaluation and implementation of the health system performance research network, Bridgepoint electronic patient reported outcome tool. | Steel Gray, Wodchis, Upshur, Cott, McKinstry, Mercer, Palen, Ramsay, Thavorn [38] | Pragmatic cluster randomized controlled trial–intervention groups using ePRO tool compared with control groups on measure of quality of life, patient experience, and cost-effectiveness; evaluating of tool. |
| 22 | 2017 | Decision-making and goal-setting in chronic disease management: baseline findings of a randomized controlled trial. | Kangovi, Mitra, Smith, Kulkarni, Turr, Huo, Glanz, Grande, Long [39] | Randomized controlled trial–patients used low-literacy aid to prioritize one of their chronic conditions and then set a goal for that condition with their primary care provider; patients created patient-driven action plans for reaching these goals. |
| 23 | 2017 | Goal-directed health care: redefining health and health care in the era of value-based care. | Mold [40] | Review |
| 24 | 2017 | Patient self-defined goals: essentials of person-centered care for serious illness. | Schellinger, Anderson, Frazer, Cain [41] | Descriptive qualitative analysis–initial inquiry to describe self-defined goals patients living with advanced heart failure, cancer, and dementia; goals were entered in electronic health record flow sheet using patients' quotes; analysis of 160 flow sheets with a deductive approach. |
| 25 | 2017 | A three-goal model for patients with multimorbidity: a qualitative approach. | Vermunt, Harmsen, Elwyn, Westert, Burgers, Rikkert, Faber [42] | Qualitative study–qualitative interviews with general practitioners and clinical geriatricians and analyzed following a thematic approach. |
| 26 | 2017 | Collaborative goal setting with elderly patients with chronic disease or multimorbidity: a systematic review. | Vermunt, Harmsen, Westert, Rikkert, Faber [17] | Systematic review based on EPOC, PRISMA and MOOSE guidelines; Pubmed, PsychInfo, CINAHL, Web of Science, Embase, Cochrane Central Register of Controlled Trials were searched systematically; eligibility criteria: 1) Randomized (cluster) controlled trials, non-randomized controlled trials, controlled before-after studies, interrupted time series or repeated measures study design; 2) Single intervention directed specifically at collaborative goal setting or health priority setting or a multifactorial intervention including these elements; 3) Study population of patients with multimorbidity or at least one chronic disease (mean age ± standard deviation (SD) incl. age 65). 4) Studies reporting on outcome measures reducible to outcomes for collaborative goal setting or health priority setting. |
| 27 | 2018 | Goal setting dynamics that facilitate or impede a client-centered approach. | Kessler, Walker, Sauvé-Schenk, Egan [24] | Conversation analysis on goal-setting conversations; purposively selected from a pilot randomized controlled trial of OPC-stroke. |
| 28 | 2018 | Development of a clinically feasible process for identifying individual health priorities. | Naik, Dindo, Van Liew, Hundt, Vo, Hernandez-Bigos, Esterson, Geda, Rosen, Blaum, Tinetti [4] | Prospective development and feasibility study–development team of patients, caregivers, clinicians using a user-centered design to develop and refine value-based patient priorities care process and medical record template; descriptive statistics and qualitative analysis of barriers and enablers. |
| 29 | 2019 | Learning from patients about patient-centeredness: a realist review: BEME guide No.60 | De Groot, Schönrock-Adema, Zwart, Damoiseaux, Jaarsma, Mol, Bombeke [43] | Realist review–realist review approach; literature search in scoping phase, deductive and inductive coding to extent rough program theory. |
| 30 | 2019 | Towards a person-centred learning health system: understanding value from the perspectives of patients and caregivers. | Kuluski, Guilcher [44] | Commentary; call to action to combine the tenets from person-centered care, value-based healthcare, and learning health systems. |
| 31 | 2019 | What is important to older people with multimorbidity and their caregivers? Identifying attributes of person centered care from the user perspective. | Kuluski, Peckham, Gill, Gagnon, Wong-Cornall, McKillop, Parsons, Sheridan [9] | Qualitative descriptive study; 1–1 interviews semi-structured interviews with 172 patients and caregivers from 9 community based primary healthcare. |
| 32 | 2019 | Putting goal-oriented patient care into practice. | Reuben, Jennings [12] | Review |
| 33 | 2019 | Setting goals with patients living with multimorbidity: qualitative analysis of general practice consultations. | Salter, Shiner, Lenaghan, Murdoch, Ford, Winterburn, Steel [23] | Qualitative analysis of general practice consultations–analysis of video recorded doctor-patient interactions; focus groups to identify core challenges of goal-setting. |

(*Continued*)

**Table 3.** (Continued)

| | | | | |
|---|---|---|---|---|
| 34 | 2019 | Tinetti, Naik, Dindo, Costello, Esterson, Geda, Rosen, Hernandez-Bigos, Smith, Ouellet, Kang, Lee, Blaum [45] | Association of patient priorities-aligned decision-making with patient outcomes and ambulatory health care burden among older adults with multiple chronic conditions. | Nonrandomized clinical trial with propensity adjustment conducted at one patient priorities care (PPC)and one usual care; participants included 163 adults aged 65 years or older who had three or more chronic conditions care for by ten primary care practitioners (PCP) trained in PPC and 203 similar patients who received usual care from 7 PCPs not trained in PPC. |
| 35 | 2020 | Eckhoff, Weiss [46] | Goal-setting: a concept analysis | Concept analysis–method of Walker and Avant, articles and book chapters were reviewed from Cumulative Index to Nursing and Allied Health Literature, Education Resources Information Center, Psych Index. |
| 36 | 2020 | Purkaple, Nagyaldi, Todd, Mold [47] | Physician's response to patient's quality-of-life goals. | Randomized controlled trial–patients were given a previsit questionnaire that included quality of life questions; physicians in the control were given no further prompting; intervention physicians were prompted to ask quality of life questions; a two-pronged design was used: prepost group where three physicians participated in 5 control and 5 intervention encounters (n = 30) and a randomized group in which 11 physicians and their patients were randomly assigned to control or intervention groups (n = 30). Video recordings of the encounters were reviewed to determine if QOL goals were mentioned and if they were utilized in decision making. |
| 37 | 2020 | Sathanpally, Sidhu, Fahami, Gillies, Kadam, Davies, Khunti, Seidu [48] | Priorities of patients with multimorbidity and of clinicians regarding treatment and health outcomes: a systematic mixed studies review. | Systematic review–MEDLINE, EMBASE, CINAHL, and Cochrane databases were searched; included studies reported health outcome and treatment priorities of adults with multimorbidity, defined as suffering from two or more chronic conditions, or of clinicians in the context of multimorbidity or both; no restriction by study design, and studies using quantitative and/ or qualitative methodologies were included. |

**Table 4. Overview of attributes.**

| Goal-oriented care is a multifaceted, dynamic and iterative process. [1,3,4,6,12,13,17,23,24,35,37,38,41,49,50] | 1.1 Goal-elicitation builds a patient-provider relationship. [1,23,24,40,47] | |
|---|---|---|
| | 1.2 Goal-oriented care entails goal-setting. | 1.2.1 Patient-provider interaction guides goal-setting.[2,4,12,13,17,23,24,30,35,37–40,42,44] |
| | | 1.2.2 Patients' needs and preferences are the foundation of SMART formulated goals. [1–4,6,10,13,23,24,26,30,32,36,39,41,44,45,47,51] |
| | | 1.2.3 Care plan is based on patients' needs and preferences. [1,3,4,6,10,12,13,17,26,28,30,33] |
| | | 1.2.4 Care is delivered according to the care plan.[1,6] |
| | 1.3 Goal-evaluation is a reflexive process. | 1.3.1 Feedback should be given to the goals. [33,49] |
| | | 1.3.2 Evaluation entails questioning how goals are being met. [12] |
| | | 1.3.3 Goals must be measurable. [13], 33) |
| 2. Goal-oriented care embraces patients' values. | 2.1 Goal-oriented care must be placed in patients' context. [3,12,26,30,37] | |
| | 2.2 Goal-oriented care must be tailored to patients' needs and preferences. [1,6,23,24,33] | |

described as a multifaceted dynamic and iterative process of care (first main attribute) underpinned by patients' values (second main attribute). For the process of goal-oriented care five sub attributes and seven descriptive items could be identified (Table 4). These attributes interact and cannot be interpreted separately.

**Goal-oriented care is a multifaceted, dynamic and iterative process.** The majority of the authors presented goal-oriented care as a stepwise approach [1,3,4,6,12,13,17,23,24,35,37,38,41,49,50]. Even though every paper defined their own approach, overall three stages could be identified: (a) goal-elicitation, (b) the actual stage of goal-setting, and (c) a reflexive goal-evaluation stage. These three stages will be further discussed.

Bernsten et al. emphasized the dynamic and iterative characteristics of the goal-oriented process of care [6]. They described that goal-oriented care entails going back and forth between the three stages [6]. From this perspective, goals are not described as an endpoint, but they can be adjusted, discarded, modified or new goals might be set [12,33]. This will be further discussed in the stage of goal-evaluation.

Overall, in the goal-oriented process of care, the patient is described as an active partner [1]. Therefore, a good communication in a continuous patient-provider relationship is described to be of utmost importance [41]. In addition, goal-oriented care should be considered as care over time rather than a one-time intervention [52]. In terms of outcomes, it is not entirely clear whether goal-oriented care should focus on (a) maintaining the status quo or (b) improving the patients' situation [12]. Although there is consensus that the care process is oriented to the current needed care rather than care needed in the future [1].

**Goal-elicitation builds a patient-provider relationship.** As described earlier, the overall analysis could identify goal-elicitation as the first stage in the process of goal-oriented care. In this first stage, providers are presumed to offer time and space to patients to tell their stories in order to work towards the patients' agenda [24]. Therefore, patients have to be ready and should be actively encouraged to tell their story. Tinetti and colleagues described this as 'the patient's state of readiness' [1]. This first stage is considered to be essential to work towards a balanced patient-provider conversation and relation. Salter et al. described this stage as a shared process between patients and providers that reinforces and further builds their relationship [23]. This specific part of the process of goal-oriented care is also described as a mean to achieve a greater level of shared understanding and mutual commitment between the patient and the provider [40]. Specific attention to the stage of goal-elicitation is described to create a supportive context for effective goal-setting in the next stage [23].

**Goal-oriented care entails goal-setting.** Besides the goal-elicitation stage, the literature identifies a goal-setting stage. Franklin and colleagues analyzed patient-provider conversations during goal-setting and concluded that the goal-setting stage serves as a mechanism to embrace patients' needs within the social context he lives in [32]. When this process is done properly, goal-setting should support the patients to continue doing what matters most to them which would help them to cope with their conditions [32]. Within this process of goal-setting different sub attributes, that are considered necessary for proper goal-setting, could be identified.

**Patient-provider interaction guides goal-setting.** The patient-provider interaction is characterized by a patient-centered approach [23] in which goals are set in collaboration [42]. Hereby, patients and providers agree on health-related goals [2,12,13,35,38,42,50,53] and find common ground [52]. Tinetti et al. described the importance of considering patients as active partners in the goal-setting process [33]. Rijken et al. mentioned that patients' goals have to be discussed in a dynamic conversation continuously taking the patients' needs, preferences, and abilities into account [2].

To facilitate a collaborative approach it is suggested that providers emphasize the patients' narratives reflecting their lived experience [40]. Besides a collaborative approach, negotiation is important and considered inevitable [4,6,23,37,49]. Lenzen et al. defined this as goal-negotiation, which involves discussion of any kind of problems, exploration of the patients' values, needs and capabilities, and deliberation on patients' goals [37]. In goal-negotiation, formulating and agreeing on a specific goal are important components [23].

Because the goal-setting process needs to be driven by patients' needs and preferences, there seems to be a general understanding to shift the focus from the provider to the patient [24]. Different authors reported various strategies to facilitate this shift. Mold stated that the shift implies that prioritization of the individual health-related goals and the amount of effort in achieving them should be made by the individual [13]. Naik et al. stated that patients are indeed encouraged to share their priorities, but adds that providers are encouraged to align their care with the patients' health priorities [4]. More recent publications talking about goal-setting describe a circular and shared process aimed at improving the balance and power differentials in the patient-provider relationship [4,39]. This balance can be improved by putting themselves in someone's shoes to understand the other's constraints [44].

**Patients' needs and preferences are the foundation to set goals.** One of the important challenges in our understanding of the concept of goal-oriented care is the lack of clear understanding on patient goals. Nearly all authors described that goals should be grounded on the patients' needs and preferences [1–4,6,23,24,32,33,39,41,47,49,52,54]. It is described that goals should be based on the context, resources and capabilities of patients [47], that they should be approved by patients [6], and that they should foremost represent what the patients want and not necessarily what the providers want [12,41]. Other authors recommended that goals should be a combination of both the patients' goals and the providers' goals which in turn is related to goal-negotiation [24,44]. In conclusion, no overall understanding on the goals could be formulated.

Besides this lack in understanding, there also seems to be ambiguity about the categorization of goals. Some authors emphasized that goals should contain core values of patients (e.g., the broader aspects that matter most to the patient) [1,4]. These goals are named as 'overarching goals' [6,12,24,41] leading to a broad description of the goal (e.g., I want to live in my own home as long as possible [1]) [6]. Others argued that these overarching goals might not be easy to work with and describe that these goals should be broken down into sub goals (e.g., I want to walk 2 blocks without shortness of breath [1]) [6]. Goals differ for each individual and will change over time [13]. Aside from overarching goals and sub goals many of the authors

mention the importance of setting SMART goals [1,6,23,24,35,46,49,50,52]. A SMART goal is created when patients and providers collaborate to untangle the goal itself, the importance of that goal is emphasized to the patient, the perceived achievability of the goal is evaluated, as well as the timing of the goal, and any supports and resources available [35]. On the meta-perspective, overarching goals are too broad to make SMART (think about the grandmother aiming to get her grandchildren from school as long as possible). Therefore they should be divided in the sub-goals (such as I need to be able to walk without being tired after 10 yards) that are specific enough to be measured.

In one of his first publications Mold brings in a specific discourse around the categorization of goals, namely that goal-oriented care should assist patients in achieving their maximum individual health potential [13], hereby making the link with health. One should however notice that health should be described from the patients' perspective; as the ability to live their life, and not as the absence of disease [1,13]. Patients' goals are oriented towards health outcome goals. Patients hope to achieve these individual health outcomes through their health care (e.g., function, social activities, and symptom relief) [1]. Health outcome goals describe activities that promote change in physical and cognitive well-being or health [36]. Naik et al. specifically relate patient goals to the care they are willing to receive and able to perform [4].

**Care plan is based on patients' needs and preferences.**   Many authors relate goal-oriented care to the construction of a care plan based on the patients' needs and preferences and specifically mention that these care plans should reflect the patients' personal goals that have been identified in the previous stage [1–3,6,12,26,28]. There is a consensus that the care plan should reflect the question: 'What matters to you?' [12,33,44,49,54]. Strategies to achieve the patients' needs and preferences should be implemented in the care plan [13]. Furthermore, Bernsten and colleagues stated that the care plan might also include an interprofessional review of the goals [6]. Therefore, it is necessary to involve all providers and preferably patients' informal caregivers and family in the whole process [3,6,17]. In case that providers are confronted with patients' goals that are out of their own scope, they could benefit from an interprofessional review as they are enabled to discuss with and hand over to other providers with the required expertise. This could improve the coordination of the care plans between the different providers and facilitate integrated care delivery [1,4,30]. To guide this interprofessional review, no specification was given about which profile would be the best fit for having the lead. Vermunt et al. (2017) illustrated this as they found variation in who (e.g., GP, nurse, practice nurse, psychological wellbeing practitioner) should contribute to goal-setting [17].

**Care delivery according to the care plan.**   Patients and providers should implement the care plan and translate it into care delivery. Although, little is known about how care should be delivered, it is evident that it must be in accordance with the care plan that is set up in the previous stage [6]. For this stage Tinetti et al. specifically mentioned to start the stage of care delivery by prioritizing on simple interventions in order to achieve one or more small goals to keep patients motivated [1]. This simple interventions could focus on the sub-goals described in previous paragraphs to eventually work towards the overarching goals.

**Goal-evaluation is a reflective process.**   The overall synthesis/analysis of the literature could identify goal-evaluation as the third and final stage in the process of goal-oriented care. For this stage authors described a dynamic and iterative process that allows reflection and feedback next to assessing whether and how goals have been met [33,49]. In this process goals can be redefined and adjusted. Possible reasons to adjust goals might be that goals have been too difficult to achieve or were no longer desired or relevant to the patients' situation [12]. Although many authors acknowledge the possibility and importance of goal adjustment, there is also discussion that goal-oriented processes of care requires that goals can be measured [13]. Steele Gray and colleagues described the importance of qualifying and quantifying the process

proceeded to achieve the goals [38]. In contrast, Salter and colleagues described that making the goals measurable could overcomplicate and distance the patient from their own goal and might therefore not be beneficial to the process of goal-oriented care [23].

**Goal-oriented care embraces patients' values.** In the previous attributes, goal-oriented care is described as a dynamic and iterative process in which two underpinning values are identified [4]. First, goal-oriented care must be placed in the patient's context and second, goal-oriented care must be tailored to the patient's needs and preferences.

**Goal-oriented care must be placed in patients' context.** The whole goal-oriented process of care starting from goal-elicitation to goal-evaluation needs to be placed in the patient's context. According to different authors this means that the process must be tailored to the patient's situation [3,12,37,54]. This does not only refer to the personal context, but also to the social and the cultural context. Therefore, this process is influenced by different contextual factors that should must be taken into account when developing the care plan [30,37].

**Goal-oriented care must be tailored to patients' needs and preferences.** When reviewing the attributes, it is clear that patients' needs and preferences form the common thread. The question 'What is the matter with the patient' must be retranslated to 'What matters to the patient?' [1,6,23,33]. This question enables patients to tell their story and open up in which they are considered to reflect on their achievements and personal agenda [24]. As a result, patients will have the feeling to be approached as a person instead of through their condition [6].

## Cases

The method of Walker and Avant prescribes that several cases should be described to illustrate the attributes defined in step 4 [20]. The first case of Joseph (Box 1) encompasses all the attributes identified in the literature and is therefore identified as a model case. To make this case more lively, each attribute and sub attribute of goal-oriented care is labelled in the box. It is a fictive example of delivering care according to the goal-oriented process of care with focus on the underpinning attributes. The second case of Ben (Box 2) is identified as an additional case as it lacks one or more of the attributes. E.g., in the case of Ben the stage of goal-evaluation is missing. This stage is needed to make adjustment and reflections according to the process of achieving the personal goals. Finally, the third case of Mary (Box 3) is an example of the opposite of goal-oriented care. This is described as a contrary case. In this case, the health care provider does not take the needs and preferences of Mary into account. The provider only thinks about convincing Mary of a healthy lifestyle which for her is not the main reason to visit her health care provider. Her main focus is on being able to play with her children.

## Antecedents

Antecedents are events or incidents that occur prior to the investigated concept. In this concept analysis, provider preparedness and patient preparedness are required to provide goal-oriented care.

In terms of provider preparedness many authors discussed the importance of training [6,7,24,28,32,42,50]. Notwithstanding that several authors [1,4,17,23,28,33,39] mentioned the importance of trained health care providers, there was a difference in the training they received (S1 File). Differences can be found in the target population reached with the training, both in monodisciplinary and interprofessional training (e.g., general practitioners [23], practice nurses [28], duration of the training (e.g., three hour [23], number of sessions [28]) and training method (e.g., role-play [33]). Thereby, the content of the training was tailored to the skills needed to carry out the intervention correctly and differ therefore in each training (S2 Table).

## Box 1. Model case of Joseph

Joseph, 68- year old suffers from diabetes, hypertension and chronic obstructive pulmonary disease. Throughout his entire working life, he was a secondary school teacher. He has been retired for three years now *(patients' context)*. Despite the fact that he is limited by his health condition, he loves spending time gardening and playing with his grandchildren *(patients' needs and preferences)*.

A few years ago he was a passionate cyclist, but his racing bike has been stored for a long time now. His friends encourage him to cycle with them on a weekly base *(patients' context)*. His wife supports this initiative and argues that this will be beneficial for his social contact *(patients' context)*. Every month Joseph visits his family doctor for a check-up. For each consultation, he prepares a list of things he wants to discuss. He has the chance to share his story in an open communication in which trust and mutual respect are key components *(goal-elicitation)*.

In his monthly check-up with his family doctor he suggests his wishes to cycle again with his friends *(patients' needs and preferences & goal-setting; interaction)*. His doctor doubts whether this will be possible and after discussion and negotiation *(goal-setting; interaction)*, they plan that he would join his friends in their weekly cycling trip but only for the first two hours *(goal-setting; foundation for SMART goal)*. The group will be asked to adapt their pace and Joseph will make sure that he does not need to return back home on his own. The doctor makes adjustments to the medication scheme according to the increased efforts Joseph will make *(goal-setting; care plan)*. He will also contact the cardiologist to inform him about the changes to the medication schema *(goal-setting; care plan)*. The family doctor and the cardiologist will collaborate in order to succeed in Joseph's goal *(goal-setting; care delivery)*. The family doctor and Joseph agree to discuss and evaluate the course after three months *(goal-evaluation; feedback & evaluation)*. It is possible to increase or decrease the intensity depending on Joseph's health state and his own preferences *(goal-evaluation; evaluation)*.

A second aspect that is discussed concerning provider preparedness focused on the personal skills of providers [1,6,17,23]. These include communication and balancing skills in which an open communication with the patient is necessary and in which an equal balance between the patient and provider is a premise [1,6,17,23]. Other defined skills were the provider's ability to listen, understand and bearing witness to the patient's story [23] and their willingness to change and learn new skills to provide care according to the goal-oriented process of care [1].

Besides provider preparedness some authors [1,12,42] specifically talked about the need of patient preparedness. Patients needed to be prepared to share their needs and preferences when entering a care relationship [1]. Some authors translate the importance of patient preparedness into patient education [1], others talked about patient guidance [11] or supporting patients in developing the skills to set personal goals [37].

## Consequences

Consequences are those events or incidents that occur as a result of a concept. For the concept of goal-oriented care, the consequences defined throughout the papers could be categorized

---

### Box 2. Additional case of Ben

Ben, a 30-year old man, was renovating a house that he bought with his girlfriend when he was diagnosed with MS *(patients' context)*. They made plans to marry next year and to make a world trip as honeymoon. These plans have been put aside due to the recent diagnosis. Although he was feeling down and did not have the energy to do anything he ended up with his physician. Initiated by the interaction and the conversation with his physician *(goal-elicitation)* he was enabled to set goals again and to look at the future *(goal-setting; interaction)*. The physician decided to discuss the things that Ben really likes to do as for example making travel plans and would make it possible to achieve his goals *(goal-setting; interaction)*. Although a plan has been devised towards Ben's goals *(goal-setting; care plan)*, there has never been an discussion whether or not the goals were achieved or required adjustments to new capabilities of Ben. For this reason the consultation Ben had with his physician could not labelled as goal-oriented care.

---

in: (a) patient-related consequences [1,3,4,24,30,49], (b) provider-related consequences [1,23,30,49], (c) care-related consequences [1,23,30] and (d) general consequences [4,6,30].

Patient-related consequences are the results for patients themselves after they received care following a goal-oriented process. A goal-directed approach could be expected to increase patient satisfaction, since the values, preferences, knowledge and opinions that each patient brought to the provider-patient relationship was more valued [40]. Also, emphasis was put on the changed way of communicating in which patients felt more freely and able to speak [3]. This led to the overall feeling of being heard, understood, respected and engaged in their care [30]. Furthermore, a goal-oriented process of care could lead to a better understanding and more in-depth knowledge of patients regarding their health, activation of patients to be more involved in their care and an increase in their overall commitment. This resulted in the increase of adherence [3]. Also Mold argued that it could contribute to a better adherence [13]. In general, the gained in-depth knowledge of patients concerning their health and a

---

### Box 3. Contrary case of Mary

Mary is a 40-year old mother of two young children and has been obese since since her childhood. Due to her weight, she has a lot of pain in her joints and is short of breath which limits her exercising capacity. Her children are looking forward to playing outside with their mother during the summer holidays. Unfortunately, she is not able to play soccer or jump on the trampoline because of the pain. The pain becomes too much for her and after long hesitation she discusses this with her physician. The only thing she wants is to play and interact with her children as painless as possible and therefore asks her physician to prescribe some medication. Her physician does not support medication, but instructs her to first strive for a healthy weight as a solution to relieve the pain. This is not aligned with the wishes of Mary who only wanted a short-term solution to be able to play with her children. In the end, she leaves the consultation room with a referral to a dietitian and sport coach.

---

better understanding of their tasks could help to improve their quality of life [3]. This was enhanced by the maximization of function and the independence patients gained [13].

For providers, goal-oriented care assisted healthcare them in their decision-making [30] and gave them the opportunity to get to know their patients better. It enhanced patient-provider collaboration [13] and contributed therefore to more job satisfaction [23].

Care-related consequences were mainly focused on reducing costs, overtreatment and fragmentation [1,23,30], since care oriented to patients' priorities would reduce tests and treatments [45]. Bernsten et al. stated also that goal-oriented care could lead to an improvement of quality of care and quality of life [6].

Although, many positive outcomes have been presented, Reuben et al. mentioned a possible downside of goal-oriented care [10]. They described that some decisions to strive for personal goals may worsen the providers' performance on aggregated health measures. For example, when a diabetic patient chooses to not follow his diet and keep on smoking, because it would be a too big lifestyle change, his HbA1c-level would not be aligned with the guidelines. Although, it could be a positive outcome from the patient perspective, it would influence the quality of care provided and the population health in a negative way.

## Empirical referents

Empirical referents provide an overview of the identified assessments tools related to the attributes aiming to make the concept measurable.

None of the papers mentioned an empirical referent to measure the entire concept of goal-oriented care. Therefore, tools have been searched for each individual sub-attribute. Examples are listed in Table 5 which gives an overview of possible tools and presents an example item presented in that tool. Listing the existing individual empirical referents might initiate the development of an overall empirical referent.

## Conclusion of the concept analysis

Fig 2 represents the overall synthesis of this concept analysis of goal-oriented care. Goal-oriented care could be described as a health care approach encompassing a multifaceted, dynamic and iterative process underpinned by the patient's context and values. The process is characterized by three stages: goal-elicitation, goal-setting and goal-evaluation in which patients' needs and preferences form the common thread. In order to be able to deliver care according to the principles of the goal-oriented care process, both providers and patients need to be prepared. In terms of the consequences of goal-oriented care literature points to the potential of goal-oriented care to improve patients' experiences and provider well-being, the potential to reduce costs and improve the overall health of the population. Furthermore, a model, a contrary and an additional case illustrated an example of goal-oriented care in practice. The empirical referents showed that it is currently not possible to measure goal-oriented care in its entirety and presented an overview of possible referents for each sub attribute. Although the literature allowed us to gain more insight into the concept of goal-oriented care, different aspects need to be further discussed.

## Discussion and conclusion

This concept analysis aimed to tackle the lack of a common understanding of goal-oriented care by identifying the attributes, antecedents and consequences using the method of Walker and Avant [20]. The overall analysis showed that a goal-oriented care generally entails three stages. Despite these three stages the process of goal-oriented care cannot be implemented as a linear protocol or checklist. Two underpinning attributes, the patient's context and the

**Table 5. Overview empirical referents.**

| Attribute | Purpose of the tool | Example of item in the assessment tool |
|---|---|---|
| Goal-elicitation | | |
| Davis Observation Code (DOC) [55] | 20-item direct observation scale for physician-patient interactions | Discussing family, medical, or social history and/ or current family functioning. |
| Goal-setting | | |
| Patient goal priority questionnaire [56] | Patient-specific measure for identification of behavioral goals and evaluation of clinically significant changes | Which activities are most important for you to manage? |
| Self-identified goals assessment [57] | 1) Helps patients to identify personally meaningful occupational goals to be addressed in therapy 2) evaluate changes levels of patient-defined success in desired occupations | Think about all of the things you want to be able to do. It might help to think about the things you did at home before you went to the hospital, and things that are hard to do now. What types of things would you like to work on or improve on in therapy before you go back home? |
| Canadian Occupational Performance Measure (COPM) [58] | Measure of a client's self-perception of occupational performance in the areas of self-care, productivity, and leisure | Semi-structured interview–discussing daily functioning and personal life. |
| Health outcome prioritization tool [59] | Tool for decision-making among older persons with multiple chronic conditions | I would like to know how important 'keeping you alive', 'maintaining independence', 'reducing or eliminating pain' and 'reducing or eliminating symptoms of dizziness, fatigue, shortness of breath' is to you. |
| Electronic Patient Reported Outcome Tool (ePRO-tool) [60] | Tool can help patients and providers to collaboratively develop healthcare goals | Goal-setting for five different areas identified as most important. |
| Goal-evaluation | | |
| Goal-attainment scale [61] | Tool to measure in which extent patients' goals have been met | Determining goal-attainment using 5-point scale. |
| Patient Assessment of Care for Chronic Conditions (PACIC) [62] | Tool to measure quality of chronic disease care | Asked to talk about my goals in caring for my condition. |
| Goal-setting evaluation tool [63] | Tool to rate the quality of goals and action plans | Does the plan identify specific actions or activities that could help to reach the goal? |
| Person's context and patient's needs and preferences | | |
| Person-centered primary care measure (PCPCM) [64] | 11-item patient-reported measure to assess primary care aspects | My doctor or practice knows me as a person/ Over time, the practice helps me to meet my goals. |
| Patient centered observation form (PCOF) [65] | Tool to help healthcare providers communicate effectively with patients | Collaborative upfront agenda setting. |

patient's needs and preferences form the common thread throughout this goal-oriented process of care. These underpinning attributes represent the philosophy of care. Goal-oriented care is a continuous interaction where you go back and forth to gain a person-centered approach (Fig 2).

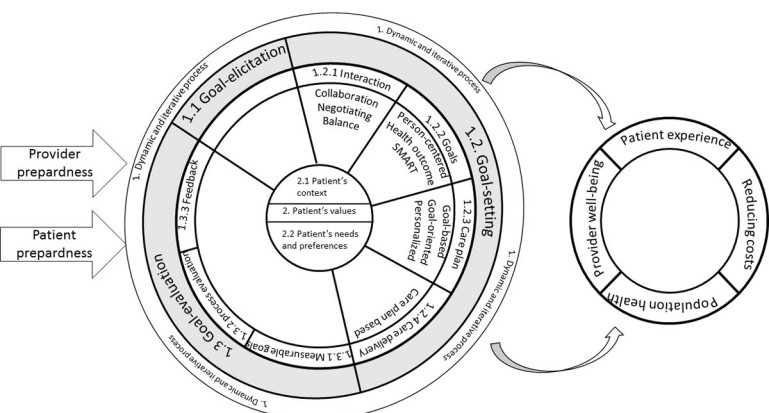

**Fig 2. Schematic representation of the antecedents, attributes and consequences.**

In the stage of goal-elicitation, greater consideration should be given to the patients' peripheral narrative reflecting their lived experiences [32]. Several authors have investigated components of goal-elicitation. Murdoch and colleagues performed a conversation analysis of patients-providers interaction during their encounters and found that eliciting the patients' understanding is an important component [66]. Ospina et al. investigated the extent to which patients' concerns are elicited across different clinical settings [67]. They concluded that providers seldom elicit the patients' agenda. This reduces the chance that providers will orient their consultation towards the specific aspects that matter to the patient [67]. One of the pre-requisites to succeed in goal-elicitation is the mutual understanding about the expectations of the consultations between patients and providers and a qualitative relationship between patients and providers [66]. The literature also mentions that patients need to have a set of skills to make appropriate health decisions and reflect on their health care choices [68]. They have to be capable to open up and tell their story [69]. It is important that patients understand the meaning of information communicated by the provider, must appreciate the consequences of the treatment options, and must reason about the information based on his or her own values and preferences [69].

Besides the stage of goal-elicitation, the stage of goal-setting was defined. One of the remaining knowledge gaps is on what kind of goals patients set. In goal-oriented care it seems important to set goals based on the patients' needs and preferences (e.g., I want to take my grandchildren to school), while in other chronic disease management programs emphasis is mainly still on health-related goals (e.g., I want the patient to walk without pain) [4]. Various work in different settings identified that patients do not necessarily have clearly defined goals for themselves [66]. Although, several authors performed research on the categorization of patients' goals. Vermunt et al. performed for example a qualitative study to develop conceptual descriptions of goal-oriented care [42]. They presented a three-level goal hierarchy containing disease- or symptom specific goals, functional goals, and fundamental goals which provides more insight in the type of goals. A second example is the distinction made by Schellinger et al. between medical, nonmedical, multiple, and global goals [41]. Not only is there ambiguity on what goals patients set, it is also not clear how goals are being set. What is clear is that patients and providers must collaborate and negotiate on which goals are important. Nevertheless, this can still cause conflicts between the patients' goals and providers' goals [26,59]. To overcome these conflicts, it is suggested to first set the patients' goals and then discuss about the medical goals, because conflicts are more likely when goals are placed on the same level [27]. It should however be noticed that setting the patients' goals on top does not legitimate full patients' responsibility over the care plan [27]. Another way to overcome these conflicts is to work with a facilitator as Naik et al. did in developing their *patients priorities identification process*. These facilitators supported patients in setting goals, choosing the most important goals to eventually communicate them with the provider [4]. Yet another strategy is to use tools to assess patient treatment priorities and preferences. Unfortunately, Mangin et al. found few relevant tools to set patients' goals [30]. They argue for the need to develop specific strategies to make patient priorities visible in the clinical record and medical-decision making [30].

Goal-evaluation was pointed out as the last stage. As presented in the results, several authors described that goals should be made measurable for evaluation [23,60]. There are some pitfalls related to goal-evaluation. Salter et al. described that not all goals lend themselves to being measured [23]. It is for example challenging to evaluate the goal 'I want to take my grandchildren from school as long as possible'. Another pitfall is that patients' goals would be simplified to what can be measured. Working towards goal-evaluation might increase the pressure on patients and providers to work in the same way as disease-specific guidelines do [70]. Especially from the perspective of patients with multimorbidity it can be questioned whether

disease-specific guidelines that are good for the disease are also good for the patient [70]. Furthermore, evidence shows that older multimorbid patients place quantitative health outcomes, such as longer survival, on a lower level of importance [70]. The focus must be on the patients' values and make healthcare more humane [40].

As mentioned for the antecedents it is important that patients and providers are prepared to work towards a goal-oriented process of care. The collaboration and co-creation between the two partners and in an interprofessional team is an important but insufficient prerequisite to succeed in providing goal-oriented care. Currently patients are not always stimulated to think about their care. They have to be stimulated to actively engage their narrative and to share their priorities. Also providers have to develop complementaty skills in which they learn to let go their own assumptions and solutions. They have to learn to integrate patients' narrative in their care plan and improve their communication skills to strengthen the mutual understanding between them [71]. Voigt et al. observed that GPs are often unaware of patients' priorities in daily life, which were in contrast with their perceived importance of patient's medical goals [71]. Training and tools could provide the guidance needed to improve the communication [1,4,17,23,28,33,39]. It could support providers in structuring the conversation, to set goals in collaboration with patients, and to align their care to those goals. Not only does goal-oriented care offers a specific approach for one-on-one interaction between patients and providers, it could also facilitate interprofessional collaboration. It gives providers from divers disciplines the opportunity to deliver care following the same principles and to focus on pursuing patients' goals [35]. Therefore training should also include the interprofessional perspective to facilitate a uniform attitude towards the patients' goals and principles of goal-oriented care in the entire team. This will potentially support providers to learn from and with each others' expertise and enable discussion between them in case that, for example, patients set goals that our out of the remit of the provider. Besides patient and provider preparedness, it could seem logical that also the system has to be prepared, but the current literature does not point to that.

In terms of the consequences of goal-oriented care, a limited number of studies have been able to demonstrate outcomes of goal-oriented care. Nonetheless, these studies showed mostly positive outcomes towards the patients, providers, health system, and overall population well-being. In that respect, goal-oriented care shows the potential to meet the components of the quadruple aim. It can be questioned if all providers experience increased satisfaction and well-being in providing goal-oriented care. Providers have to learn to cope with another way of delivering care. For example, a changed medication scheme as described in Josephs' case in order to work towards patients' goals. This goes against their basic principles to strive for the best possible health status including a comprehensive medication scheme. Besides that the provider well-being can be questioned, Blom et al. also contradicted the positive results for the health care system. They did not find a beneficial effect in health care use and costs when using a proactive, goal-oriented, integrated care model [28].

One of the reasons of the limited number of effectiveness studies of goal-oriented care is the lack of empirical referents. The concept must still undergo the transition towards an evaluable concept. Boyd et al. argue for measures for quality of care needed by older persons with multimorbidity as the current clinical guidelines have undesirable effects for this population [51]. Goal-oriented care is identified by Etz and colleagues as one of the main constructs when developing a new comprehensive measure of high-value aspects of primary care, however they did not mention how it has to be done [72]. Further Young et al. described outcome goals as a main construct when differentiating processes and outcomes for primary care and divided it further in goal-clarity for multimorbidity, goal-clarity for unique patient priorities and goal timing [73]. It is clear that in order to gain more insight in the consequences of goal-oriented care further research must primarily focus on how goal-oriented care is provided and can be

supported. In order to investigate the potential benefits of goal-oriented care, research also needs to work on developing indicators of the goal-oriented process of care.

## Strengths, limitations, and recommendations

The method of Walker and Avant provides a rigorous and systematic approach to refine the concept of goal-oriented care through the existing literature. A concept analysis is an exploration of an evolving concept which will need to be enriched by new knowledge. Therefore, it is influenced by contextual factors and must undergo adjustments to new implications and new insights based on further research. Since there is no specification given by Walker and Avant on how to conduct the literature review, we followed the guidelines from a scoping review as described by Levac (2010) [21]. The iterative process of adding new articles following the snowballing method is one of the strengths compared to other types of reviews. In this concept analysis, this led to a larger number of articles than the original search. A possible explanation for this might be that goal-oriented care was covered by synonyms or similar concepts that were not covered by the original search. Despite the systematic approach, a concept analysis does not comprise a quality assessment of the literature. However, it seemed to be an appropriate method to provide the knowledge needed to understand the different components of goal-oriented care in its entirety. The literature that was included in this study were only English written and peer reviewed. It would however be interesting to add also non-English literature to be able to capture more differences (e.g., cultural differences).

The literature search identified both original research papers and position papers. Some original research papers [3,4,23,38,41] evaluated goal-oriented care in clinical practice. These papers identified and described goal-oriented care as a stepwise intervention. Position papers [1,12,13,35,37] mostly described components of goal-oriented care rather than such a stepwise approach. The combination of both types gave more insight in the broad components of goal-oriented care.

This concept analysis could also be considered as a preliminary step to facilitate further research. One of the knowledge gaps revealed in this concept analysis is the lack of knowledge on what patients' goals are set, how goal-oriented care is delivered, and how it is best put into practice in both one-on-one interactions between patients and providers and in interprofessional collaboration. Regarding patients it is important to gain more insight in how they are preferably prepared for discussing their personal goals. In addition, the list of empirical referents made clear that a golden standard to evaluate goal-oriented care is missing. Initiating the development of an evaluation method could enable future intervention studies to gain more insight in the consequences of goal-oriented care and to make results comparable. Increasing insights from effective goal-oriented care could highlight its multiple benefits towards providers and policy makers. These results might also inform the healthcare system in which resources they need to facilitate goal-oriented care. A following step will first be to discuss these theoretical insights with patients and providers and deepen this information with insights from practices. Then, when goal-oriented care is well understood, a critical review can be set up to perform in-depth comparison between other concepts and frameworks. At this moment, we have (unfortunately) insufficient information to do this.

Goal-oriented care shows the potential to be a way forward for patients with chronic conditions and multimorbidity. However, further research is needed to translate the current knowledge on the concept of goal-oriented care into a tangible workflow process of care that entails the three stages. This workflow should consists of tools to prepare patients and providers to offer goal-oriented care. This could contribute to finding a common ground in the goals and implementing goal-oriented care in practice.

## Conclusion

This concept analysis aimed to translate the concept of goal-oriented care into a common understanding so providers can better understand and use this concept in clinical practice. The various literature on goal-oriented care, based on position and original research papers, showed a stepwise approach of three stages. Overall, the underpinning attributes of patients' context and patients' values form a philosophy of care to which the process must be reflected. Furthermore, both patients and the providers need to develop new skills in order to rethink the way care is provided. Patients must therefore be enabled to open up and reflect on their own agenda. Providers instead must learn to let go their own assumptions and solutions and communicate with their patients in a more balanced context. Based on the literature goal-oriented care shows the potential to improve patients' experience by listening to their needs and preferences, improve providers' well-being by the feeling of more satisfaction and reduce health care costs. Goal-oriented care could answer the challenges patients face with multiple care processes by initiating interprofessional collaboration. However, further research must focus on what and how goals are set, the translation of these findings into a workflow and must initiate the development of an evaluation method in order to investigate the effects of goal-oriented care processes on patients, providers and the health care system.

## Supporting information

**S1 Table. Overview preliminary version attributes.**
(PDF)

**S2 Table. Overview training.**
(PDF)

**S1 File. Preferred Reporting Items for Systematic reviews and Meta-Analyses extension for Scoping Reviews (PRISMA-ScR) checklist.**
(PDF)

## Acknowledgments

We are grateful for the partnership with the Primary Care Academy (academie-eerstelijn.be) and want to thank the King Baudouin Foundation and Fund Daniel De Coninck for the opportunity they offer us for conducting research and have impact on the primary care of Flanders, Belgium. The consortium of the Primary Care Academy consists of: Lead author: Roy Remmen–roy.remmen@uantwerpen.be—Department of Primary Care and Interdisciplinary Care, Faculty of Medicine and Health Sciences. University of Antwerp. Antwerp. Belgium; Emily Verté—Department of Primary Care and Interdisciplinary Care, Faculty of Medicine and Health Sciences. University of Antwerp. Antwerp. Belgium, Department of Family Medicine and Chronic Care, Faculty of Medicine and Pharmacy. Vrije Universiteit Brussel. Brussel. Belgium; Muhammed Mustafa Sirimsi—Centre for research and innovation in care, Faculty of Medicine and Health Sciences. University of Antwerp. Antwerp. Belgium; Peter Van Bogaert—Workforce Management and Outcomes Research in Care, Faculty of Medicine and Health Sciences. University of Antwerp. Belgium; Hans De Loof—Laboratory of Physio pharmacology, Faculty of Pharmaceutical Biomedical and Veterinary Sciences. University of Antwerp. Belgium; Kris Van den Broeck—Department of Primary Care and Interdisciplinary Care, Faculty of Medicine and Health Sciences. University of Antwerp. Antwerp. Belgium.; Sibyl Anthierens—Department of Primary Care and Interdisciplinary Care, Faculty of Medicine and Health Sciences. University of Antwerp. Antwerp. Belgium; Ine Huybrechts

—Department of Primary Care and Interdisciplinary Care, Faculty of Medicine and Health Sciences. University of Antwerp. Antwerp. Belgium.; Peter Raeymaeckers—Department of Sociology, Faculty of Social Sciences, Faculty of Social Sciences. University of Antwerp. Belgium; Veerle Buffel- Department of Sociology; centre for population, family and health, Faculty of Social Sciences. University of Antwerp. Belgium.; Dirk Devroey- Department of Family Medicine and Chronic Care, Faculty of Medicine and Pharmacy. Vrije Universiteit Brussel. Brussel.; Bert Aertgeerts—Academic Centre for General Practice, Faculty of Medicine. KU Leuven. Leuven, Department of Public Health and Primary Care, Faculty of Medicine, KU Leuven. Leuven; Birgitte Schoenmakers—Department of Public Health and Primary Care, Faculty of Medicine, KU Leuven. Leuven. Belgium; Lotte Timmermans—Department of Public Health and Primary Care, Faculty of Medicine, KU Leuven. Leuven. Belgium.; Veerle Foulon—Department of Pharmaceutical and Pharmacological Sciences, Faculty Pharmaceutical Sciences. KU Leuven. Leuven. Belgium.; Anja Declerq—LUCAS-Centre for Care Research and Consultancy, Faculty of Social Sciences. KU Leuven. Leuven. Belgium.; Nick Verhaeghe— Research Group Social and Economic Policy and Social Inclusion, Research Institute for Work and Society. KU Leuven. Belgium.; Dominique Van de Velde Department of Rehabilitation Sciences, Occupational Therapy. Faculty of Medicine and Health Sciences. University of Ghent. Belgium., Department of Occupational Therapy. Artevelde University of Applied Sciences. Ghent. Belgium.; Pauline Boeckxstaens—Department of Public Health and Primary Care, Faculty of Medicine and Health sciences. University of Ghent. Belgium.; An De Sutter -Department of Public Health and Primary Care, Faculty of Medicine and Health sciences. University of Ghent. Belgium.; Patricia De Vriendt—Department of Rehabilitation Sciences, Occupational Therapy. Faculty of Medicine and Health Sciences. University of Ghent. Belgium., Frailty in Ageing (FRIA) Research Group, Department of Gerontology and Mental Health and Wellbeing (MENT) research group, Faculty of Medicine and Pharmacy. Vrije Universiteit. Brussels. Belgium., Department of Occupational Therapy. Artevelde University of Applied Sciences. Ghent. Belgium.; Lies Lahousse—Department of Bioanalysis, Faculty of Pharmaceutical Sciences, Ghent University. Ghent. Belgium.; Peter Pype—Department of Public Health and Primary Care, Faculty of Medicine and Health sciences. University of Ghent. Belgium., End-of-Life Care Research Group, Faculty of Medicine and Health Sciences. Vrije Universiteit Brussel and Ghent University. Ghent. Belgium.; Dagje Boeykens- Department of Rehabilitation Sciences, Occupational Therapy. Faculty of Medicine and Health Sciences. University of Ghent. Belgium., Department of Public Health and Primary Care, Faculty of Medicine and Health sciences. University of Ghent. Belgium.; Ann Van Hecke—Department of Public Health and Primary Care, Faculty of Medicine and Health sciences. University of Ghent. Belgium., University Centre of Nursing and Midwifery, Faculty of Medicine and Health Sciences. University of Ghent. Belgium.; Peter Decat—Department of Public Health and Primary Care, Faculty of Medicine and Health sciences. University of Ghent. Belgium.; Rudi Roose—Department of Social Work and Social Pedagogy, Faculty of Psychology and Educational Sciences. University Ghent. Belgium.; Sandra Martin—Expertise Centre Health Innovation. University College Leuven-Limburg. Leuven. Belgium.; Erica Rutten—Expertise Centre Health Innovation. University College Leuven-Limburg. Leuven. Belgium.; Sam Pless —Expertise Centre Health Innovation. University College Leuven-Limburg. Leuven. Belgium.; Vanessa Gauwe—Department of Occupational Therapy. Artevelde University of Applied Sciences. Ghent. Belgium.; Didier Reynaert- E-QUAL, University College of Applied Sciences Ghent. Ghent. Belgium.; Leen Van Landschoot—Department of Nursing, University of Applied Sciences Ghent. Ghent. Belgium.; Maja Lopez Hartmann—Department of Welfare and Health, Karel de Grote University of Applied Sciences and Arts. Antwerp. Belgium.; Tony Claeys- LiveLab, VIVES University of Applied Sciences. Kortrijk. Belgium.; Hilde

Vandenhoudt—LiCalab, Thomas University of Applied Sciences. Turnhout. Belgium.; Kristel De Vliegher—Department of Nursing–homecare, White-Yellow Cross. Brussels. Belgium.; Susanne Op de Beeck—Flemish Patient Platform. Heverlee. Belgium.

## Author Contributions

**Conceptualization:** Dagje Boeykens, Pauline Boeckxstaens, An De Sutter, Lies Lahousse, Peter Pype, Patricia De Vriendt, Dominique Van de Velde.

**Formal analysis:** Dagje Boeykens, Pauline Boeckxstaens, Patricia De Vriendt, Dominique Van de Velde.

**Funding acquisition:** Pauline Boeckxstaens, An De Sutter, Lies Lahousse, Peter Pype, Patricia De Vriendt, Dominique Van de Velde.

**Investigation:** Dagje Boeykens.

**Methodology:** Pauline Boeckxstaens, Patricia De Vriendt, Dominique Van de Velde.

**Supervision:** Pauline Boeckxstaens, Patricia De Vriendt, Dominique Van de Velde.

**Validation:** Pauline Boeckxstaens, An De Sutter, Lies Lahousse, Peter Pype, Patricia De Vriendt, Dominique Van de Velde.

**Writing – original draft:** Dagje Boeykens.

**Writing – review & editing:** Pauline Boeckxstaens, An De Sutter, Lies Lahousse, Peter Pype, Patricia De Vriendt, Dominique Van de Velde.

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
