## [Decision Letter · Decision Letter 0]

4 May 2021

PONE-D-20-30664

Goal-oriented care in primary care: a concept analysis

PLOS ONE

Dear Dr. Boeykens,

Thank you for submitting your manuscript to PLOS ONE. After careful consideration, we feel that it has merit but does not fully meet PLOS ONE’s publication criteria as it currently stands. Therefore, we invite you to submit a revised version of the manuscript that addresses the points raised during the review process.

We look forward to receiving your revised manuscript.

Kind regards,

Greg Irving, FRCGP MPH PhD

Academic Editor

PLOS ONE

Journal Requirements:

2. During the internal evaluation of the manuscript, we feel that this study fits within the scope of a Scoping Review. As such we please consider modifying your title to specify this. For instance "Goal-oriented care in primary care: a scoping review and concept analysis

In addition please provide a PRIMSA flow chart as Figure 1, list of studies as Table 1, and a completed PRISMA-Scr checklist as Supporting File

Finally, please include in your Methods section the date ranges over which you conducted the literature search

3. One of the noted authors is a group or consortium Primary Care Academy. In addition to naming the author group, please list the individual authors and affiliations within this group in the acknowledgments section of your manuscript. Please also indicate clearly a lead author for this group along with a contact email address.

Additional Editor Comments:

Dear author

We have now received comments from four reviewers. I would be grateful if you could address the issues they raise and then resubmit the manuscript.

Best wishes

Dr Greg Irving

Reviewers' comments:

Reviewer's Responses to Questions

**Comments to the Author**

1. Is the manuscript technically sound, and do the data support the conclusions?

Reviewer #1: Yes

Reviewer #2: Yes

Reviewer #3: Yes

Reviewer #4: Yes

2. Has the statistical analysis been performed appropriately and rigorously? 

Reviewer #1: N/A

Reviewer #2: N/A

Reviewer #3: N/A

Reviewer #4: N/A

3. Have the authors made all data underlying the findings in their manuscript fully available?

Reviewer #1: Yes

Reviewer #2: Yes

Reviewer #3: Yes

Reviewer #4: Yes

4. Is the manuscript presented in an intelligible fashion and written in standard English?

Reviewer #1: Yes

Reviewer #2: Yes

Reviewer #3: Yes

Reviewer #4: Yes

5. Review Comments to the Author

Reviewer #1: Thank you for sending me this paper to review. It provides a very thorough concept analysis of goal oriented care, reviewing a range of studies upon this topic. Overall it succeeds in providing a clear definition in the results that will be a useful characterisation on which to base other studies, particularly given the often overlapping terms and definitions available on this topic. The methods appear to be sound and clearly outlined. However, the paper would benefit from further elaboration or consideration of issues in places, including:

Title:

- no mention is made of multimorbidity, a key concept within the paper. It should be added in.

Introduction:

- an assumption is made throughout that goal oriented care is likely to be better - can the authors provide any effectiveness data related to this topic? There are a number of trials in multimorbidity in which a goal setting approach is used. Likewise in the abstract intro.

- disease-specific care is positioned as opposite to goal oriented care. However, within a number of chronic disease management programmes, goal setting plays a large part (although the patient centredness of this may be debatable). perhaps the authors need to more clearly distinguish between goal setting in care and goal-oriented care, which seems to be broader in their analysis.

- Likewise, other related concepts such as shared decision making and patient centred care are only briefly touched upon and in either the intro or discussion or both need to be discussed as to how they relate to goal oriented care.

- primary care is seen as the main focus of this paper, which makes sense, but there is little detail on collaboration and how goal oriented care would fit with collaborative approaches and whose responsibility in the primary care team it could/should be

-

Methods:

- I appreciate the literature searches are iterative but it would be useful to know the range of dates searched or at the very least the date of the most recent search, in order to contextualise the point at which this was done

- L139 - 'confirmed the first results' unclear what this means - no new studies found? or no new concepts identified?

- In the inclusion criteria, it is unclear how much of a focus was needed on goal oriented care to be included - was there a minimum level of discussion or characterisation required? How many authors determined it had sufficient focus? To me this would seem to be difficult to judge.

- clarify that all paper types were included

- L153 - "a chronic condition"? This paper is focused on multimorbidity so surely it should be >1?

- papers included were English only. It would be interesting to know how many papers were in another language as it seems like that there may be cultural differences that could not be picked up?

-

Results:

- flow chart too low res to view

- table 2 is clear but would appreciate adding further details on methods, plus perhaps contribution it made to the concept analysis (could add in numbers from table 3 to indicate where contributed to?). Some study methods are blank? I don't see the relevance of journal.

- table 3 is clear and results largely clear

- p18 patient's needs and preferences... within this section I wondered if any of the papers had picked up on expectation management, which would seem to be an important part of the process which is not really considered here. Additionally on page 19 I wondered what the implications were when people's goals went beyond the remit of healthcare professionals in terms of their aims or their barriers and facilitators (e.g. finances, caring responsibilities) and whether the responsibility should be on the HCP to address these kind of issues as part of holistic care or the patient as part of their own self-management? This could also be picked up on the discussion and implications.

- it is somewhat unclear who should be leading/doing the care plan. Whilst an interprofessional approach is emphasised, there is little detail on whether a specific role (e.g. care coordinator or similar) would be required for this process to take place.

- there's also little differentiation throughout as to whether the goals set are meant to be actioned by the patient or providers, which is an important consideration when considering measurement and review and goal level.

- throughout it would be good to know when a paper is quoted, the type of data/expertise it is coming from and whether these characterisations are consistent across methods (e.g. whether characterised in the same way from a detailed conversational analysis perspective vs an overview paper)

- The case boxes are cut off so I can't comment on them. Box 3 states 'her physician does not allow' - does this mean he is preventing Mary from travelling, which does not seem like something he would be able to, or is this perhaps a slight miswording?

- in the patient preparedness part, could an conclusions be drawn about the best ways to prepare patients?

Discussion

- should be a bit wider not just in terms of the concept but how it fits in more widely. There is overall a lack of discussion around the issues of provider-provider collaboration (for example frequently care plans however holistically developed are not shared across providers, particularly those with different IT systems) and healthcare professional time (which would seem to be the major barrier)

Couple of English corrections needed throughout: L109 chronical should be chronically, L211 independency should be independence

Reviewer #2: To the authors

Thank you for your great work.

This article touched on the really important point of family medicine.

I hope these comments improve your article.

Major concerns

#1. As the authors mentioned, there are similar concepts with Goal-oriented care. For example, Patient-Centered Clinical Method, Value-Based Practice or Expert Generalist Practice have proposed similar frameworks in primary care settings. I think the authors need to mention the difference between Goal-oriented care and the other concepts and why the authors chose Goal-oriented care in the Background and Discussion section. This is important for readers who are not familiar with Goal-oriented care.

References

Patient-Centered Clinical Method: Patient-Centered Medicine: Transforming the Clinical Method

Value-Based Practice: https://www.cambridge.org/core/series/valuesbased-practice/62B7FBB835241FBFEBD1016EB2DA7860

Expert Generalist Practice: https://bmcfampract.biomedcentral.com/articles/10.1186/1471-2296-14-112

#2 Please clarify how to “analyze” the included articles to define attributes in step 4. This can be helpful to understand the process of emerging codes for the readers who are not familiar with the concept analysis.

Reviewer #3: Dear Authors,

Thank you for your work. I enjoyed reading your paper about the concept of goal-oriented care.

At first I was expecting a more interpretative approach rather than a structured concept analysis , but I imagine that many will find this an interesting take. I have some comments that may help to improve the paper further.

Regarding the methods I assume there is a good reason why related concepts including patient-centred care, shared decision making and value based medicine were not included in the search string, but could this be argued more clearly?

I can’t read the figure 1, the resolution is too low.

On page 18: what is the confusion in the paragraph above exactly? Different aspects are mentioned, but what is the tension?

Regarding the results: Isn’t there literature that defines goal oriented by stating what it is ‘not’?

Cases: The cases are not fully readable in the version of the paper that I had access to, but I think you need to work a bit on them to make them more believable or better: find actual cases. Eg. case 1: for most CVD / DM patients cycling is actually recommended and case 3: obesity is hardly a barrier to travel. In reality I guess that most often patient’s goals are not dismissed but are simply not discussed (for many reasons).

Page 24: was there nothing about systems preparedness? If so, please that this was not found

In the discussion I was expecting a juxtaposition of the goal oriented concept with related concepts in primary care, including generalism, holism, patient-centredness, value based healthcare, shared decision making, patient participation, EBM (Sacket!) etc. How is it different? You could be more critical: is goal oriented care actually a better concept and if so why?

Perhaps also provide a discussion on tensions with contrasting frameworks and systems such as transactionalism, care standards, P4P, neoliberal economics and budgetting (tension of providing efficient care for many).

I would also like to see what was missing in the literature / what was not mentioned? E.g. as I mentioned above:system preparedness (as opposed to patient and provider preparedness). Anything else?

Hopes this helps and I’m happy to review any future adjusted version if needs be.

Reviewer #4: Dear Colleagues

Thank you for inviting me to review this. Please accept it has taken me longer to respond than I would have liked. It has been a busy time. This is an important topic for healthcare in general, not just primary care. You have presented the review of the literature in systematic way and it is easy to read. I have added comments, mostly minor, to the PDF of the paper for consideration. The term 'empirical referent' is hard to understand, although I can see you have described it - in order to bring it to life I wonder if the table with the examples could provide more details. As I could not read the full case studies I cannot comment- but these are very important to bring the paper to life. I wondered if a bit more discussion is needed about the risks of when patient and providers goals do not align, and approaches needed to align these.

I would recommend publication as it is an important piece of work to enable further research in this field and support the application of goal-oriented care by patients and providers.

Best wishes

Luisa

6. PLOS authors have the option to publish the peer review history of their article (what does this mean?). If published, this will include your full peer review and any attached files.

Reviewer #1: No

Reviewer #2: **Yes: **Makoto Kaneko

Reviewer #3: No

Reviewer #4: No

---

## [Author Response · Author response to Decision Letter 0]

2 Jun 2021

Dear editor

Dear reviewers

We appreciate your extensive and constructive feedback on our manuscript, it added a lot of value.

We aimed to respond to all of your remarks by point to point answers and changes in the manuscript. Please, find the changes in the 'response to the reviewers letter'.

Hopefully they will meet you expectations, otherwise we will make further adjustments. 

Sincerely

---

## [Decision Letter · Decision Letter 1]

9 Dec 2021

PONE-D-20-30664R1Goal-oriented care for patients with chronic conditions or multimorbidity in primary care: a scoping review and concept analysisPLOS ONE

Dear Dr. Boeykens,

Thank you for submitting your manuscript to PLOS ONE. After careful consideration, we feel that it has merit but does not fully meet PLOS ONE’s publication criteria as it currently stands. Therefore, we invite you to submit a revised version of the manuscript that addresses the points raised during the review process.

We look forward to receiving your revised manuscript.

Kind regards,

Filipe Prazeres, MD, MSc, Ph.D.

Academic Editor

PLOS ONE

Journal Requirements:

Reviewers' comments:

Reviewer's Responses to Questions

**Comments to the Author**

1. If the authors have adequately addressed your comments raised in a previous round of review and you feel that this manuscript is now acceptable for publication, you may indicate that here to bypass the “Comments to the Author” section, enter your conflict of interest statement in the “Confidential to Editor” section, and submit your "Accept" recommendation.

Reviewer #2: All comments have been addressed

Reviewer #4: All comments have been addressed

2. Is the manuscript technically sound, and do the data support the conclusions?

Reviewer #2: Yes

Reviewer #4: Yes

3. Has the statistical analysis been performed appropriately and rigorously? 

Reviewer #2: N/A

Reviewer #4: N/A

4. Have the authors made all data underlying the findings in their manuscript fully available?

Reviewer #2: Yes

Reviewer #4: Yes

5. Is the manuscript presented in an intelligible fashion and written in standard English?

Reviewer #2: Yes

Reviewer #4: Yes

6. Review Comments to the Author

Reviewer #2: Thank you for providing opportunity to review the manuscript.

I really enjoyed reading the article.

Reviewer #4: Dear Authors

Many thanks for doing such a great job of responding to all the comments made by the reviewers. I am happy to recommend for publication with minor correction made to the English used (comments made on attached PDF). However, I also feel strongly that your 3 case examples need to be further strengthened with detail and reference back to the attributes (or absence of them) of goal-oriented care in table 4. It is quite a hard concept to grasp and the case studies are your opportunity to bring it to life - but I don't think they currently fully do.

Luisa

7. PLOS authors have the option to publish the peer review history of their article (what does this mean?). If published, this will include your full peer review and any attached files.

Reviewer #2: **Yes: **Makoto Kaneko

Reviewer #4: **Yes: **Dr Luisa M Pettigrew

---

## [Author Response · Author response to Decision Letter 1]

13 Dec 2021

Dear Editor

Dear Reviewers

Thank you for the positive feedback and pointing to minor comments to improve our manuscript. 

Please find our answers in our rebutall letter.

Sincerely

Dagje Boeykens, on behalf of the co-authors

---

## [Editor Report · Decision Letter 2]

7 Jan 2022

Goal-oriented care for patients with chronic conditions or multimorbidity in primary care: a scoping review and concept analysis

PONE-D-20-30664R2

Dear Dr. Boeykens,

We’re pleased to inform you that your manuscript has been judged scientifically suitable for publication and will be formally accepted for publication once it meets all outstanding technical requirements.

Kind regards,

Filipe Prazeres, MD, MSc, Ph.D.

Academic Editor

PLOS ONE
---

## [Editor Report · Acceptance letter]

12 Jan 2022

PONE-D-20-30664R2 

Goal-oriented care for patients with chronic conditions or multimorbidity in primary care: a scoping review and concept analysis. 

Dear Dr. Boeykens:

I'm pleased to inform you that your manuscript has been deemed suitable for publication in PLOS ONE. Congratulations! Your manuscript is now with our production department. 

Kind regards, 

on behalf of

Prof. Filipe Prazeres 

Academic Editor

PLOS ONE